# Distinction of *Paramecium* strains by a combination method of RAPD analysis and multiplex PCR

Sonoko Matsumoto[1☉], Kenta Watanabe[1,2☉], Hiroko Kiyota[2], Masato Tachibana[3], Takashi Shimizu[1,2], Masahisa Watarai[1,2]*

1 Joint Graduate School of Veterinary Medicine, Yamaguchi University, Yamaguchi, Japan, 2 Joint Faculty of Veterinary Medicine, Laboratory of Veterinary Public Health, Yamaguchi University, Yamaguchi, Japan, 3 Joint Faculty of Veterinary Medicine, National BioResource Project Paramecium, Yamaguchi University, Yamaguchi, Japan

☉ These authors contributed equally to this work.
* watarai@yamaguchi-u.ac.jp

**Data Availability Statement:** All relevant data are within the paper and its Supporting Information files.

## Abstract

*Paramecium* is employed as a valuable model organism in various research fields since a large number of strains with different characteristics of size, morphology, degree of aging, and type of conjugation can be obtained. It is necessary to determine a method for the classification and simple identification of strains to increase their utility as a research tool. This study attempted to establish a polymerase chain reaction (PCR)-based method to differentiate strains of the same species. Genomic DNA was purified from several strains of *P. caudatum*, *P. tetraurelia*, and *P. bursaria* used for comparison by the random amplified polymorphic DNA (RAPD)-PCR method. In *P. tetraurelia* and *P. bursaria*, it was sufficiently possible to distinguish specific strains depending on the pattern of random primers and amplification characteristics. For the classification of *P. caudatum*, based on the sequence data obtained by RAPD-PCR analysis, 5 specific primer sets were designed and a multiplex PCR method was developed. The comparative analysis of 2 standard strains, 12 recommended strains, and 12 other strains of *P. caudatum* provided by the National BioResource Project was conducted, and specific strains were identified. This multiplex PCR method would be an effective tool for the simple identification of environmental isolates or the management of *Paramecium* strains.

## Introduction

*Paramecium* is a single-cell free-living eukaryote who lives in freshwaters worldwide, such as ponds and lakes [1]. The artificially cultivation is also easy, and it is considered a model organism for many aspects of eukaryotic biology [2,3]. For example, its high motility is useful in cell biology as a model for studying cell motility [4,5], and its unique and complex reproduction process, including genome duplication and rearrangement, is directly related to its utilization value as a model for sexual reproduction or inheritance analysis [6,7]. *Paramecium* is also a potential host that tolerates the intracellular symbiosis of different organisms [8]. *Holospora* spp., gram-negative α-proteobacteria, are well known to be maintained within *Paramecium* nuclei as an obligate symbiont [8,9]. *Paramecium bursaria*, which establishes an intracellular

**Funding:** This study was supported in part by the Japan Society for the Promotion of Science Grant-in-Aid for Scientific Research (C) and (B) under grant nos. 19K06383 and 17H03914 to K.W. and M.W. The funders had no role in study design, data collection and analysis, decision to publish, or preparation of the manuscript.

**Competing interests:** The authors have declared that no competing interests exist.

symbiotic relationship with green alga *Chlorella*, is widely studied as a suitable model system to elucidate secondary symbiosis [10–12]. *Paramecium* is also employed as a protist host model for environmental bacteria, including pathogenic bacteria, and has been reported for its symbiotic mechanism [13–15].

Erstwhile, the classification of *Paramecium* spp. was based on morphological observations, but nowadays, with the development of genetic techniques, the classification based on genomic information, such as 18S rRNA, internal transcribed spacer (ITS), and mitochondrial cytochrome oxidase subunit 1 genes are widely used, as practiced in other species [16–18]. 18S rRNA is widely studied as it is a multicopy gene in many species, including *Paramecium*, making it easy to detect. Compared to 18S rRNA, the ITS region has higher interspecies polymorphism even among closely related species, increasing specificity in detection [19]. The ITS region is useful as an identification marker sequence that can distinguish closely related species [20]. However, it may be insufficient to distinguish between genetically related strains of the same species or classify new environmental isolates since genomic information registered in public databases is limited compared to the total number of *Paramecium* strains that have been identified. Besides, one of the unique phenotypes of *Paramecium* is the presence of syngens. The conjugation of *Paramecium* only occurs among the same syngens, and in some cases are recognized as a cryptic species. However, genes involved in the determination of syngen, whether or not they are present, have not yet been identified, nor has a method for genetically identifying them been established.

Random amplified polymorphic DNA (RAPD) analysis is one of the classical molecular biological methods to assess genetic diversity by analyzing the homology or polymorphism of DNA sequence [21–23]. The major advantage of RAPD method is that the achievements of identification or discrimination are obtained quickly and easily without previous details about the genomic DNA of the organism. Due to these advantages, this analysis has been applied to the identification or comparative analysis of *Paramecium* strains [24,25]. We have also previously applied this RAPD method to distinguish between several strains of *Paramecium* or determine if there are specific patterns among strains with a common syngen [26]. To briefly distinguish *Paramecium* spp. whose genomic information of each strain is poor and morphological distinction is difficult, the RAPD method is a suitable approach. However, the RAPD method also has a disadvantage: the results may vary greatly depending on the polymerase chain reaction (PCR) conditions and the quality of the sample genome. In other words, there is a gap in reproducibility. It was confirmed that a small amount of genome of other organisms (e.g., symbionts of *Paramecium* or bacteria fed as a feed of *Paramecium* remaining in the culture medium) contaminating the purified *Paramecium* genome could affect the results [26]. In addition, since *Chlorella*, the symbiont in *P. bursaria*, is a eukaryote as well as the host *Paramecium*, it is necessary to consider the possibility that its presence or contamination of the genome could affect the results more than bacteria.

This study searched for gene regions that can discriminate between strains of the same species of *Paramecium* based on the results obtained by the RAPD method and investigated the establishment of a convenient method to distinguish between strains. Several strains of various *Paramecium* spp. from the National BioResource Project (NBRP) were used, and a novel PCR tool was applied for strains identification.

## Materials and methods

### *Paramecium* strains

All strains of *Paramecium caudatum*, *P. tetraurelia*, and *P. bursaria* were obtained from the Symbiosis Laboratory, Yamaguchi University, supported by the NBRP (http://nbrpcms.nig.ac.jp/paramecium/?lang=en). All strains used in this study are listed in Table 1.

**Table 1. List of *Paramecium* strains used in this study.**

| Species | No. | Strain Name | NBRP ID | Syngen | Mating Type | note |
|---|---|---|---|---|---|---|
| *P. caudatum* | 1 | dYDRM-3E | PC032039A | 3 | E | recommended strain |
| | 2 | dYDRM-3O | PC031040A | 3 | O | recommended strain |
| | 3 | G3-402 | PC031044A | 3 | O | recommended strain |
| | 4 | dOW-4E | PC042005A | 4 | E | recommended strain |
| | 5 | dOW-4O | PC041006A | 4 | O | recommended strain |
| | 6 | dCRT-5E | PC052001A | 5 | E | recommended strain |
| | 7 | dCRT-5O | PC051002A | 5 | O | recommended strain |
| | 8 | YDRM-6E | PC062131A | 6 | E | recommended strain |
| | 9 | YDRM-6O | PC061132A | 6 | O | recommended strain |
| | 10 | SBK2019-12E | PC122022A | 12 | E | recommended strain |
| | 11 | SBK2019-12O | PC121226A | 12 | O | recommended strain |
| | 12 | My43C3d | PC121015B | 12 | O | recommended strain, [27] |
| | 13 | dKNZ-12E | PC122029A | 12 | E | standard strain |
| | 14 | dKNZ-12O | PC121031A | 12 | O | standard strain |
| | 15 | Myn92 | PC012002A | 1 | E | |
| | 16 | BAT-CIA3 | PC012001A | 1 | E | |
| | 17 | Ai102 | PC011016A | 1 | O | |
| | 18 | Mmn64 | PC011011A | 1 | O | |
| | 19 | YR1504-2 | PC062012A | 2 | E | |
| | 20 | YDRM20 | PC032036A | 3 | E | |
| | 21 | YDRM46 | PC032037A | 3 | E | |
| | 22 | TAZ0462 | PC032004A | 3 | E | |
| | 23 | RB-1 | PC042001A | 4 | E | |
| | 24 | YR1504-6 | PC061016A | 6 | O | |
| | 25 | YDRM28 | PC062134A | 6 | O | |
| | 26 | SBK2019-3b1 | PC122105A | 12 | E | |
| *P. tetraurelia* | 1 | st110-1a | PA041001A | - | O | recommended strain |
| | 2 | st110-1b | PA042002A | - | E | recommended strain |
| | 3 | rie-1 | PA042018A | - | E | recommended strain |
| | 4 | rie-2 | PA041019A | - | O | recommended strain |
| | 5 | SSZ1 | PA042017A | - | E | recommended strain |
| | 6 | KMA21 | PA041022A | - | O | recommended strain |
| | 7 | ds4-2 (VIII) | PA042004A | - | E | standard strain |
| | 8 | ds4-2 (VII) | PA041003A | - | O | standard strain |
| | 9 | 51 | PA040011A | - | E | standard strain |
| *P. bursaria* | 1 | YKK10g | PB031015A | B1 or R3 | I | recommended strain |
| | 2 | YKK10w | PB031016A | B1 or R3 | I | recommended strain |
| | 3 | Dd1g | PB032001A | B1 or R3 | II | recommended strain |
| | 4 | KM2g | PB031002A | B1 or R3 | II | recommended strain |
| | 5 | YDS1g | PB032031A | B1 or R3 | II | recommended strain |
| | 6 | YDS1w | PB032061A | B1 or R3 | II | recommended strain |
| | 7 | YKK3g | PB033046A | B1 or R3 | III | recommended strain |
| | 8 | YKK3w | PB033049A | B1 or R3 | III | recommended strain |
| | 9 | HA1g | PB034004A | B1 or R3 | IV | recommended strain |
| | 10 | HA1w | PB034007A | B1 or R3 | IV | recommended strain |
| | 11 | Yad1g1N | PB031010B | B1 or R3 | I | standard strain, [11] |

## Isolation of genomic DNA from *Paramecium*

Genomic DNA was isolated from 15 mL *Paramecium* cells culture (including ~10,000 cells) using a DNA extraction kit (Qiagen, Venlo, The Netherlands) according to the manufacturer's instructions. Cells were washed with phosphate-buffered saline twice to remove extracellular bacteria before the DNA extraction by the kit. The DNA concentration in each sample was measured by Nanodrop-ND 2000 (Thermo Scientific, USA) for qualitative and quantitative analyses. The DNA samples were frozen at −30°C until use.

## RAPD analysis

RAPD analysis was performed as described previously [26] with some modifications. In brief, RAPD-PCR was carried out in a 10 μL reaction mixture consisting of 5 μL PCR master mix (Tks Gflex DNA Polymerase; Takara Bio, Shiga, Japan), 1 μL primer (10 μM), 1 μL template DNA (adjusted to 10 ng/μL), and 3 μL nuclease-free water. The PCR program consisted of the first 4 cycles at a denaturation temperature of 94°C for 5 min, followed by annealing of the primer at 35°C for 5 min and final elongation at 72°C for 2 min. The subsequent 36 cycles consisted of denaturation at 94°C for 1 min, primer annealing at 40°C for 1 min, and elongation at 72°C for 2 min. A last elongation step was extended to 5 min at 72°C. The fragments were separated by electrophoresing at 100 V for 30 min on 2% agarose gel. The gels were stained with ethidium bromide and visualized using the gel imaging system (ATTO, Tokyo, Japan). All RAPD-PCR analyses were repeated at least thrice to confirm the reproducibility of the band patterns. The primers used in this RAPD analysis are listed in Table 2.

## DNA sequencing

After electrophoresis on a 2% agarose gel, RAPD-PCR products (several single bands that differed in size and presence or absence depending on the strain) were extracted using the QIAEx II Gel Extraction Kit (Qiagen) according to the manufacturer's instructions and used as template DNA. PCR was performed using KOD-Plus-Neo polymerase (Toyobo, Osaka, Japan)

**Table 2. Primers used in this study.**

| Primer name | Sequence (5'-3') | Expected size of the PCR product (bp) | References |
|---|---|---|---|
| 02 | GCAGGATACG | - | [24,28] |
| 03 | CTGCGATACC | - | |
| 04 | GCAGAGAAGG | - | |
| 05 | CTAGCTCTGG | - | |
| 09 | CGCTGTTACC | - | |
| Pc_1 F | GGATATATGTACATGTGAAT | 1487 | This study |
| Pc_1 R | ATAAAACAGTTGATGTTTGG | | This study |
| Pc_2 F | TAGCTCTGGACAGAATAGTG | 1078 | This study |
| Pc_2 R | TAGCTCTGGGATAAAGGAAA | | This study |
| Pc_3 F | GCAGGATACGGCAATGTAAA | 689 | This study |
| Pc_3 R | GCAGGATACGTATCAACACA | | This study |
| Pc_4 F | GCAGAGAAGGGTAATACTTC | 554 | This study |
| Pc_4 R | GCAGAGAAGGATGTAAGAGC | | This study |
| Pc_5 F | GCTGTTACCAATTAGACATC | 503 | This study |
| Pc_5 R | CGCTGTTACCAAGTATGAAG | | This study |
| 18S F | GATGGTAGTGTATTGGAC | 618 | [29] |
| 18S R | TTGGCAAATGCTTTCGC | | |

with primers designed for cloning the PCR products into *Eco*RI cleaved pCold TF DNA vector (Takara Bio). Cloning was performed using an In-Fusion HD Cloning Kit (Takara Bio). The inserted DNA was amplified by pCold-F1 and pCold-R primers. These PCR products were purified and submitted to sequencing analysis using an ABI 3130 Genetic Analyzer (Applied Biosystems, Foster City, CA) with the Big Dye Terminator version 3.1 Cycle Sequencing Kit (Thermo Fisher Scientific, Waltham, MA).

### Conventional and multiplex PCR for distinguishing *P. caudatum* strains

Conventional PCR was carried out using KOD-Plus-Neo polymerase. The reaction conditions were 2 min at 94˚C, followed by 30 cycles of 10 s denaturation at 94˚C, 30 s annealing at 60˚C, 30 s extension at 68˚C, and 7 min final extension at 68˚C. The fragments were separated by electrophoresing at 100 V for 30 min on 1% agarose gel. Multiplex PCR was carried out using the Multiplex PCR Assay Kit version 2 (Takara Bio). The reaction conditions were 1 min at 94˚C, followed by 30 cycles of 30 s denaturation at 94˚C, 45 s annealing at 57˚C, 45 s extension at 72˚C, and 10 min final extension at 72˚C. The fragments were separated by electrophoresing at 100 V for 30 min on 3% agarose gel. Gel staining and visualization were carried out as described above. Based on the sequence information (S1 File), specific primers were designed to identify *P. caudatum* strains. 18S rRNA was used as a control gene to check the quality of the template DNA. The primers used in this assay are also listed in Table 2.

## Results

### Comparative analysis of *P. caudatum*, *P. tetraurelia*, and *P. bursaria* using the RAPD method

The NBRP designate 6 strains of *P. caudatum*, *P. bursaria*, and *P. tetraurelia* as standard strains and the other 53 strains of the 24 species as recommended strains. Therefore, we first decided to conduct a comparative study using these designated strains. After the comparative study using these designated strains and ten random primer sets from a previous study [24,26,28], differences in band patterns were found between strains using only the random primer-02, -03, and -05 (Figs 1–3). In particular, results using primer-05 for *P. caudatum* (Fig 1C), primer-02 for *P. tetraurelia* (Fig 2A), and primer-03 for *P. bursaria* (Fig 3B) showed different patterns for each strain and were most effective in roughly identifying strains. Especially, No. 9 (51), one of the standard strains of *P. tetraurelia*, and No. 11 (Yad1g1N), a standard strain of *P. bursaria*, showed different numbers and patterns of bands compared to other strains (Figs 2 and 3). In contrast, some strains of *P. caudatum* showed quite similar band patterns among the recommended and standard strains. It was slightly more difficult to distinguish recommended and standard strains of *P. caudatum* by the RAPD method alone than *P. tetraurelia* and *P. bursaria*. The RAPD method did not reveal any specific band patterns that could distinguish between syngens and mating types in *P. caudatum*. In addition, *P. bursaria* showed similar band patterns in strains with *Chlorella* (Nos. 1, 3, 4, 5, 7, 9, and 11) and without *Chlorella* (Nos. 2, 6, 8, and 10). No distinct bands specific to strains with *Chlorella* were observed in any results. As expected, the same parental strain of *P. bursaria* with and without *Chlorella*, symbiotic green alga, showed very similar band patterns, for example, No.7 (YKK3g) and No.8 (YKK3w), or No.9 (HA1g) and No.10 (HA1w). However, there were cases where the band pattern was similar, although the strains were different, as in No. 3 (Dd1g) and No. 4 (KM2g).

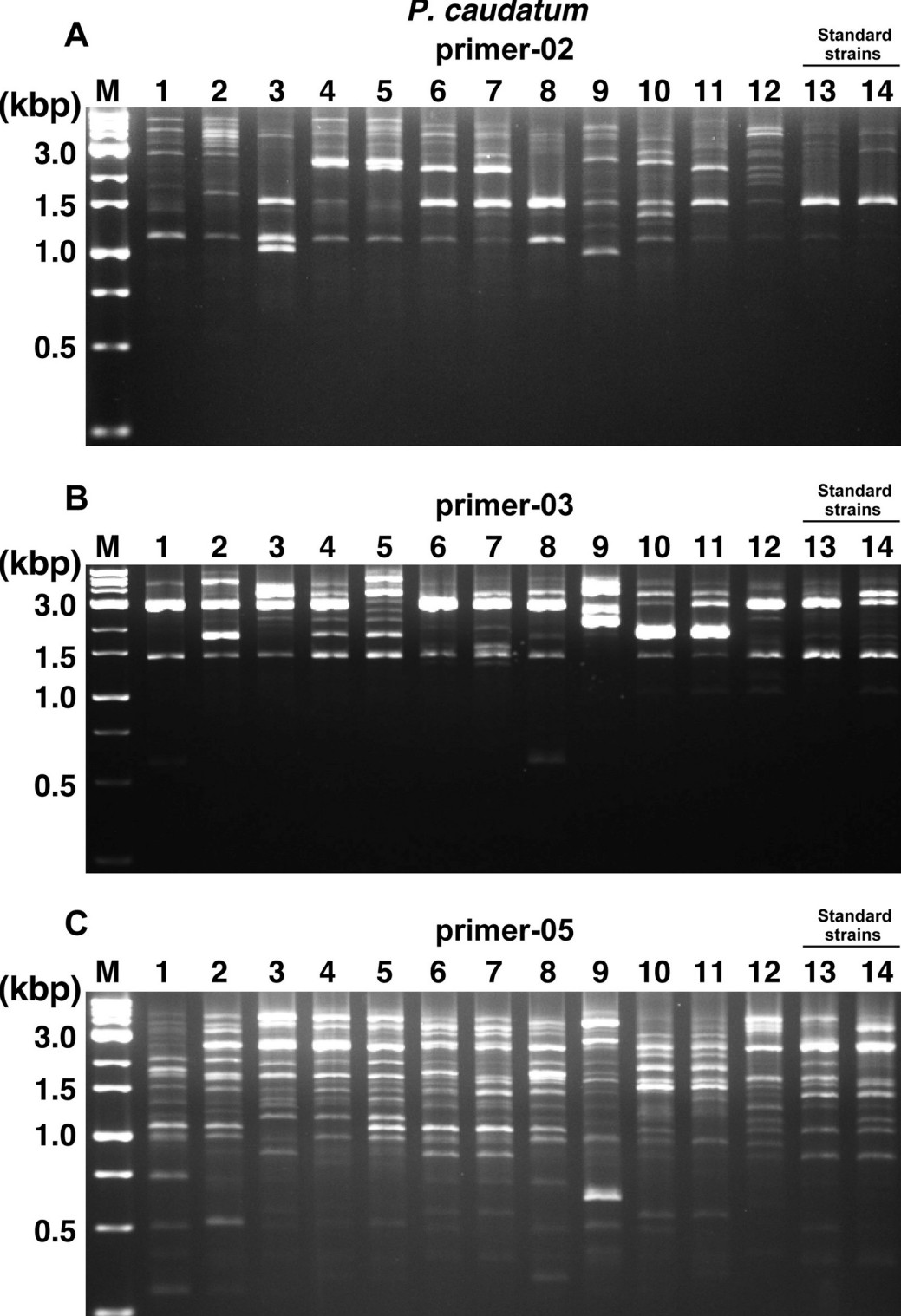

**Fig 1. RAPD analysis using template DNA from *P. caudatum* strains.** RAPD analysis using template DNA from 12 recommended strains (Nos. 1–12) and 2 standard strains (Nos. 13 and 14) of *P. caudatum*. The random primers used in each reaction are presented at the top of the figures (A, primer-02. B, primer-03. C, primer-05). M, weight marker.

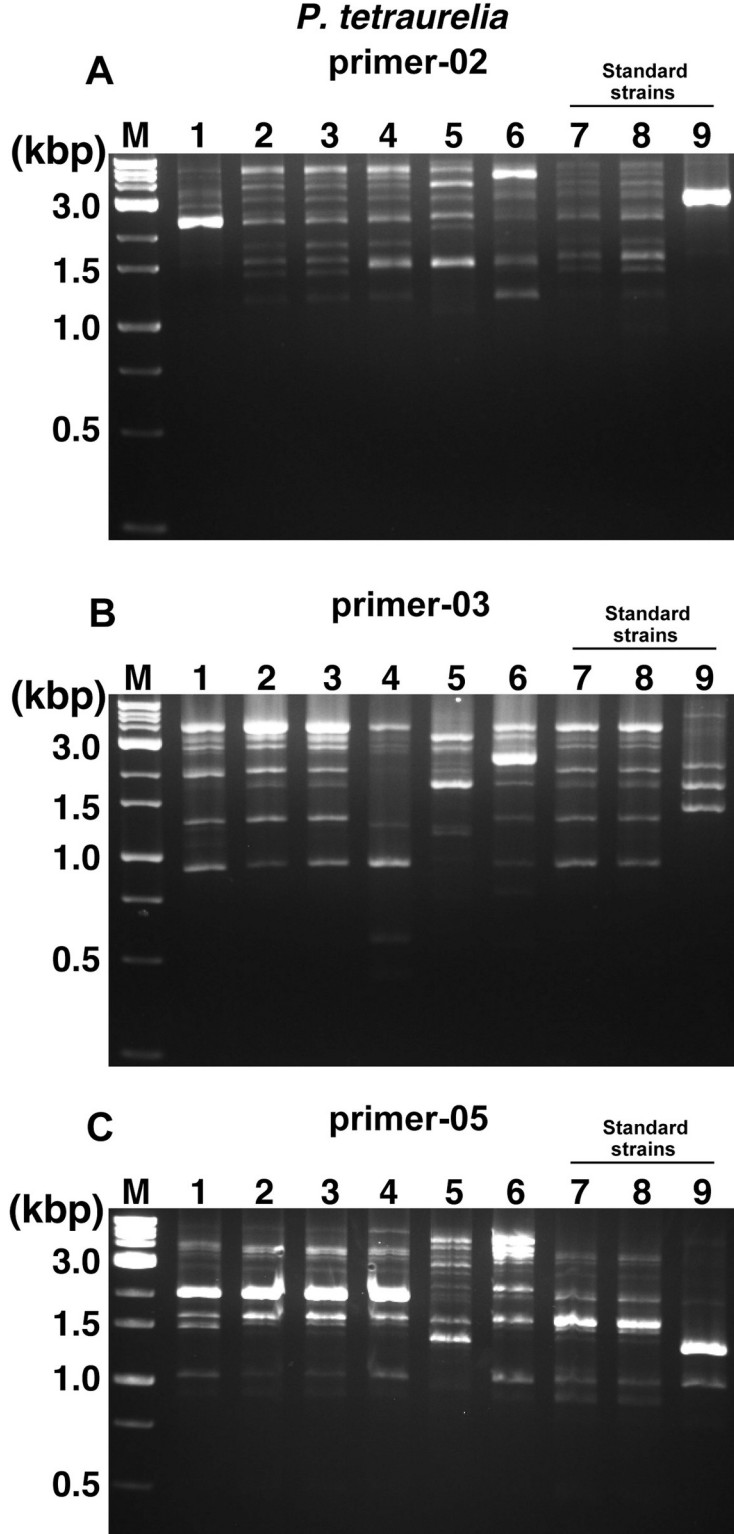

**Fig 2. RAPD analysis using template DNA from *P. tetraurelia* strains.** RAPD analysis using template DNA from 6 recommended strains (Nos. 1–6) and 3 standard strains (Nos. 7–9) of *P. tetraurelia*. The random primers used in each reaction are presented at the top of the figures (A, primer-02. B, primer-03. C, primer-05). M, weight marker.

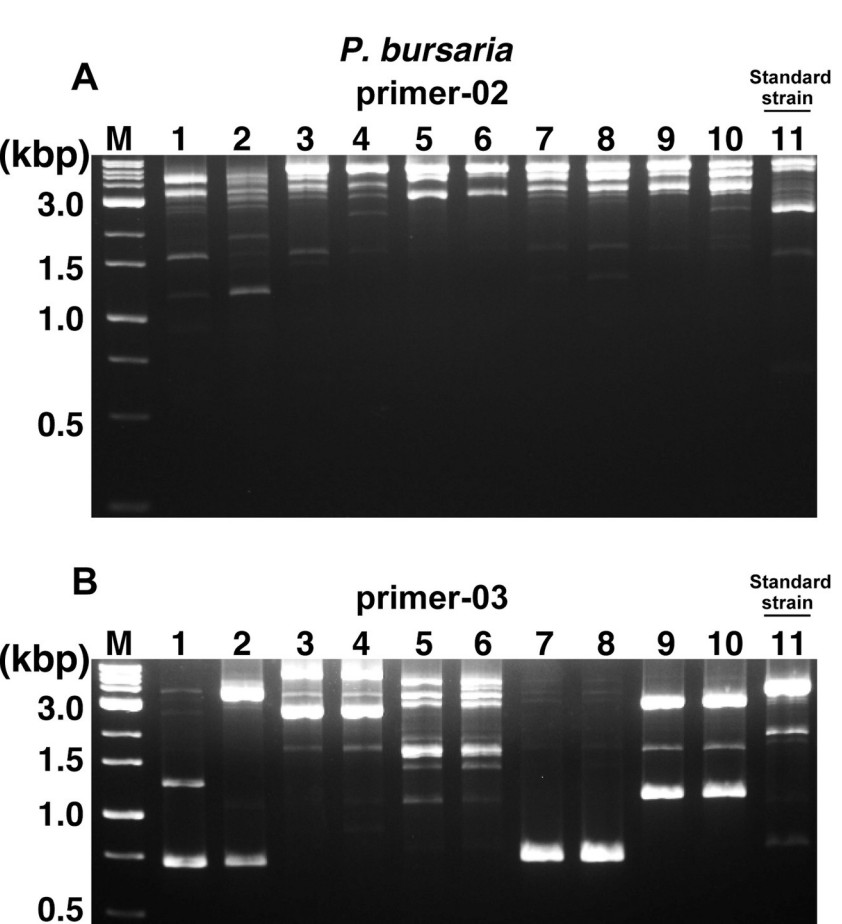

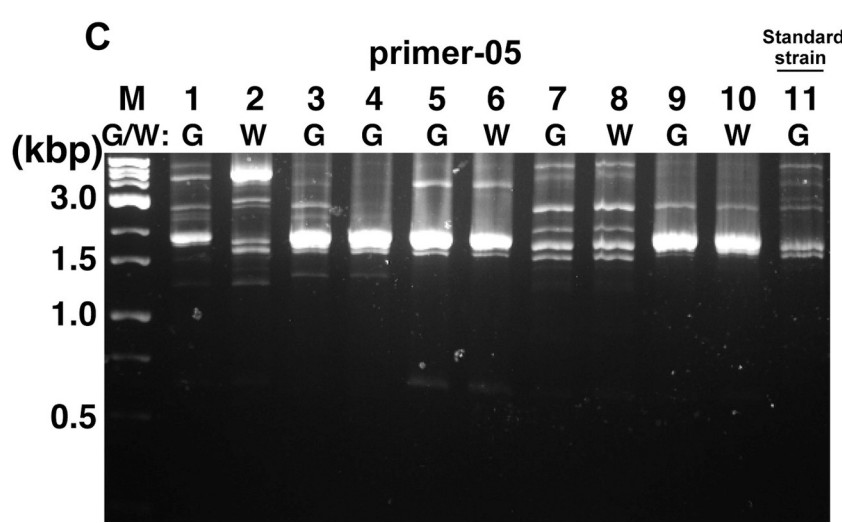

**Fig 3. RAPD analysis using template DNA from *P. bursaria* strains.** RAPD analysis using template DNA from 10 recommended strains (Nos. 1–10) and a standard strain (No. 11) of *P. bursaria*. G, strain with *Chlorella*. W, strain without *Chlorella*. The random primers used in each reaction are presented at the top of the figures (A, primer-02. B, primer-03. C, primer-05). M, weight marker.

## Distinguishing *P. caudatum* strains depending on PCR for five candidate sequences selected from RAPD-PCR results

This study attempted to develop a method for more strictly distinguishing strains in *P. caudatum*. First, specific primers were designed based on the sequence information (S1 File) obtained from RAPD-PCR products (Fig 4A). Next, comparative analysis was performed

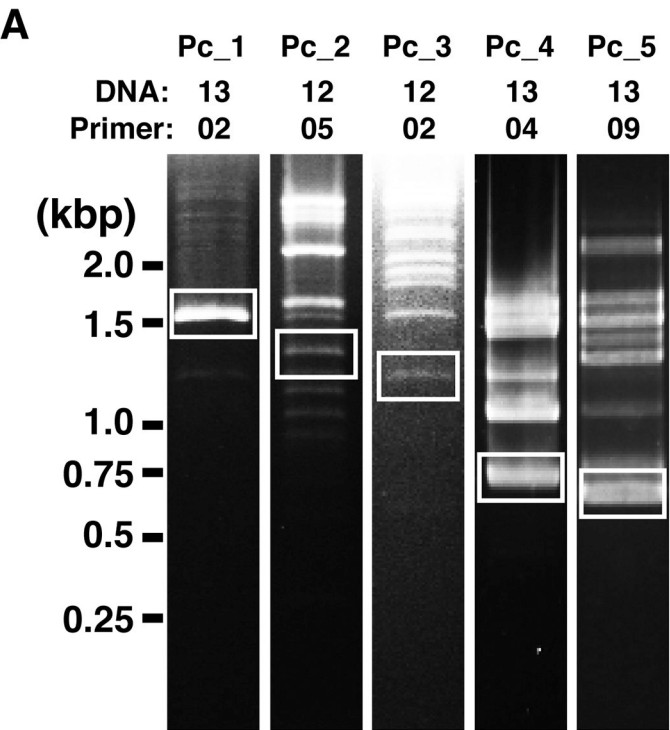

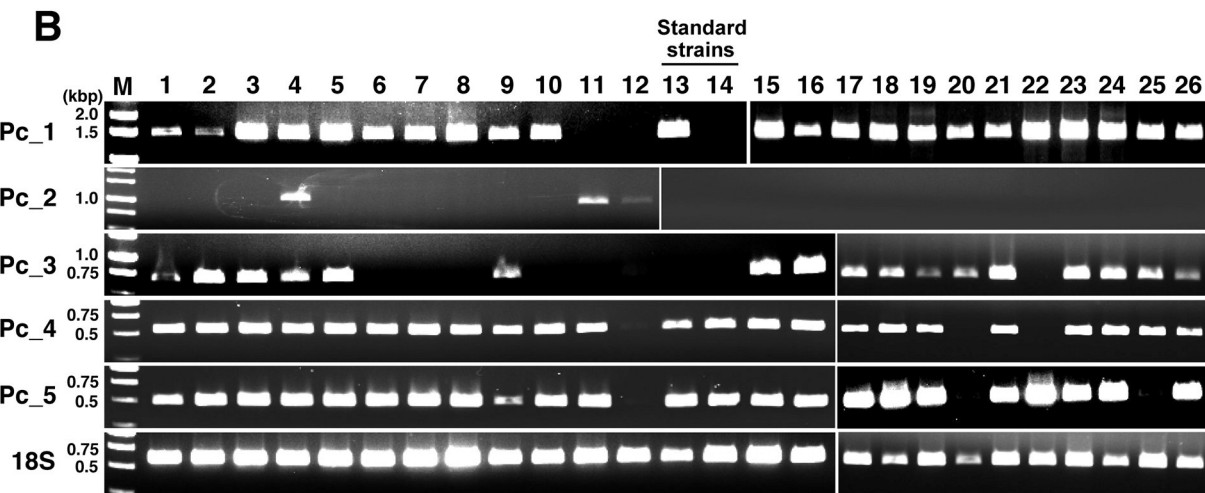

**Fig 4. Selecting the target gene region to identify strains based on the RAPD-PCR and PCR results for these target sequences.** (A) RAPD-PCR was conducted individually using template DNA from No. 12 or 13 of *P. caudatum* and random primer-02, 04, 05, and 09, individually. Single bands which were selected and submitted to sequencing analysis were surrounded by white boxes. (B) PCR results using template DNA from 26 strains of *P. caudatum* are shown. Pc_1 to Pc_5 and 18S indicate target region names. 18S, 18S rRNA gene. M, weight marker.

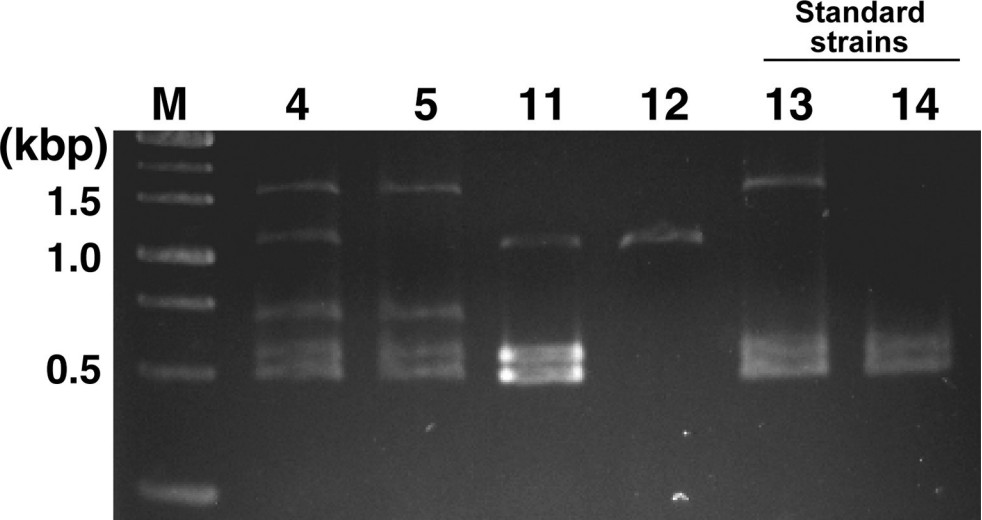

**Fig 5. Multiplex PCR.** Multiplex PCR results using template DNA from 6 strains (Nos. 4, 5, and 11–14) of *P. caudatum*. M, weight marker.

using these primers on 26 strains, including the standard and recommended strains used in the RAPD method described above, and another strain of *P. caudatum*, which is not designated as the standard or recommended strain in the NBRP. As a result, in the PCRs targeting the five regions named as Pc-1 to Pc-5, the patterns of the bands were differentiated among each strain, and it was possible to distinguish some of the strains, including the standard strain (No. 14; dKNZ-12O), by comprehensively determining the results of these five individual PCRs (Fig 4B). This study also successfully distinguished strains (e.g., Nos. 4 and 5 and Nos. 10 and 11) that were difficult to distinguish clearly since the band patterns were similar in all cases using random primers in the RAPD method.

### Examination of the application to multiplex PCR

Finally, this study investigated a simple method to distinguish *P. caudatum* strains by multiplex PCR. Although the PCR reagents used in Fig 4 had low reproducibility of the results, as some bands did not appear, the results were very reproducible, reflecting very well the PCR results performed individually targeting Pc_1 to PC_5 in Fig 4 using multiplex PCR-specific reagents (Fig 5). A maximum of five single bands (No. 4) was obtained with the predicted size, and no smearing or nonspecific bands due to primer-dimer were observed. As a result, by conducting this multiplex PCR once, it was possible to easily distinguish between several standard and recommended strains of *P. caudatum*.

### Discussion

Although the identification of *Paramecium* species has largely relied on methods based on morphological observations, this is not a completely effective method; indeed, the species are frequently changed and reclassified [30–32]. Recently, genomic information, such as 18S rRNA, ITS, and mitochondrial cytochrome oxidase subunit 1 genes has become available, and isolation and identification of *Paramecium* strains based on this information are being carried out [16–18]. Once the genome information of all *Paramecium* species is analyzed, a more appropriate method of species and strain identification will be established based on this information. Whole-genome sequencing is proceeding in several strains of *Paramecium* [27,33],

but its utilization is still very limited considering the cost and time. In this respect, the method established in this study is useful as a simple strain identification method for studies using multiple *Paramecium* strains or for identifying new isolates from the environment.

The RAPD method is a useful approach to identify and distinguish various species [21–23], including the identification of *Paramecium* strains. However, one of the disadvantages that must be considered is the reproducibility of the tests and the quality of the template DNA that may affect the results. By investigating the design of random primers and reaction conditions, it is expected to improve the outcomes. However, the suitability of the RAPD analysis for each target species is likely to be variable, since it was more difficult to detect differences in band patterns of *P. caudatum* (Fig 1) than in those of *P. tetraurelia* (Fig 2) or *P. bursaria* (Fig 3) in RAPD analysis performed under the same conditions. A previous study also reported the importance of considering the contamination of the fed bacterial genomes or symbiont genomes when applying this RAPD method to the analysis of *Paramecium* [26]. In this study, RAPD-PCR products were extracted from agarose gels to design primers specific for *P. caudatum*, and sequence analysis was performed (Fig 4A). Results showed that most PCR products were amplified from *Paramecium* DNA, including Pc_1 to Pc_5, also used in the following analysis. The results included that the PCR product was amplified from the genomic DNA of a bacteria (*Enterobacter aerogenes*) fed to *Paramecium*, but the rate was extremely low with only one sample out of 16 samples extracted from gel. Therefore, it would be reasonable to conclude that the band patterns observed in the RAPD method under the method and conditions described in this study are mainly genomic products of *Paramecium*.

The sequences (Pc_1–Pc_5) used in the primer design for *P. caudatum*-specific PCR (Figs 4B and 5) were analyzed for sequence homology in the database. However, all of them corresponded to genes of unknown function and their surrounding regions in *P. caudatum* and did not match the sequences of genes whose specific functions were identified (S1 File). This study would reveal genes involved in phenotypic determinations, such as syngen or mating types, also important as classification factors for *Paramecium* strains, but no such findings were obtained. Although several reports have referred to genes involved in determining mating types of *Paramecium* [34–39], genetic information on syngens of *Paramecium* is relatively lacking. Thus, the general method of identifying syngens remains deeply dependent on mating tests. As whole-genome sequences of many *Paramecium* strains are developed in the future, and more information on the function of each gene is revealed, the relationship between target DNA sequences employed in this study and the diversity of the strains will be clarified, leading to the identification of novel genes that determine syngens or mating types.

The multiplex PCR method used in this study was only applied to the analysis of *P. caudatum*. Still, it is highly possible that the method can be improved to simplify the distinction and identification of other *Paramecium* strains, including *P. tetraurelia* and *P. bursaria*, in a similar manner. This study did not attempt to perform the method because there were not enough strains of these two species to determine the utility compared to *P. caudatum*. It is important to continue to examine the utility and generality of this method by preparing more strains of *Paramecium* species other than *P. caudatum*. There may be some strains to which the present analysis method cannot be applied in such investigations. In particular, because the purified genome DNA from *P. bursaria* strains that maintain symbiont *Chlorella* will certainly be a mix of the genome of *Chlorella* and that of the host *P. bursaria*, it is necessary to consider this effect in the RAPD-PCR process. It is also important to consider the existence of endosymbiotic bacteria, such as *Holospora* and *Legionella*, when studying *P. caudatum* strains that maintain them [8,13]. It is necessary to ensure species specificity by combining multiplex PCR as in this method; at the same time, it is important to modify this method to the most appropriate one by changing the target sequence for each *Paramecium* species or strain for use.

## Conclusion

Although RAPD is a simple and rapid method for strain identification, the problem of reproducibility and the effect of the presence of DNA other than that of the target organism must be considered when identifying *Paramecium* strains. The multiplex PCR method applying the results of the RAPD method reported in this study is considerably more reliable than the RAPD method in terms of reproducibility. Multiplex PCR is also a method that can disregard unexpected contamination of DNA from nontarget organisms. In this study, we could provide a specific method for distinguishing the standard strains of several *Paramecium* species (S1 Table). Future studies should examine whether there will be any discrepancies or discrimination problems when this method is applied to the identification and differentiation of more strains of *Paramecium*.

## Supporting information

**S1 File. Result of DNA sequencing.**
(PDF)

**S1 Table. Quick identification chart for each standard strain.**
(PDF)

**S1 Raw images.**
(PDF)

## Acknowledgments

We acknowledge the technical expertise of The DNA Core Facility of the Center for Gene Research, Yamaguchi University, supported by a grant-in-aid from the Ministry of Education, Science, Sports and Culture of Japan.

## Author Contributions

**Conceptualization:** Sonoko Matsumoto, Kenta Watanabe, Masahisa Watarai.

**Data curation:** Sonoko Matsumoto, Kenta Watanabe, Hiroko Kiyota, Masahisa Watarai.

**Formal analysis:** Sonoko Matsumoto, Kenta Watanabe, Masato Tachibana, Takashi Shimizu, Masahisa Watarai.

**Funding acquisition:** Kenta Watanabe.

**Methodology:** Masato Tachibana.

**Resources:** Masato Tachibana.

**Writing – original draft:** Kenta Watanabe.

**Writing – review & editing:** Kenta Watanabe, Masahisa Watarai.

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
