## [Decision Letter · Decision Letter 0]

27 Jan 2022

PONE-D-21-36346Distinction of Paramecium strains by a combination method of RAPD analysis and multiplex PCRPLOS ONE

Dear Dr. Watarai,

Thank you for submitting your manuscript to PLOS ONE. After careful consideration, we feel that it has merit but does not fully meet PLOS ONE’s publication criteria as it currently stands. Therefore, we invite you to submit a revised version of the manuscript that addresses the points raised during the review process.

We look forward to receiving your revised manuscript.

Kind regards,

Maria Stefania Latrofa

Academic Editor

PLOS ONE

Journal Requirements:

“This study was supported in part by the Japan Society for the Promotion of Science Grant-in-Aid for Scientific Research (C) and (B) under grant nos. 19K06383 and 17H03914 to K.W. and M.W.”

We note that you have provided funding information within the Acknowledgements Section. Please note that funding information should not appear in the Acknowledgments section or other areas of your manuscript. We will only publish funding information present in the Funding Statement section of the online submission form.

“This study was supported in part by the Japan Society for the Promotion of Science Grant-in-Aid for Scientific Research (C) and (B) under grant nos. 19K06383 and 17H03914 to K.W. and M.W. The funders had no role in study design, data collection and analysis, decision to publish, or preparation of the manuscript.”

Reviewers' comments:

Reviewer's Responses to Questions

**Comments to the Author**

1. Is the manuscript technically sound, and do the data support the conclusions?

Reviewer #1: Partly

Reviewer #2: Yes

Reviewer #3: Yes

2. Has the statistical analysis been performed appropriately and rigorously? 

Reviewer #1: N/A

Reviewer #2: Yes

Reviewer #3: N/A

3. Have the authors made all data underlying the findings in their manuscript fully available?

Reviewer #1: Yes

Reviewer #2: Yes

Reviewer #3: No

4. Is the manuscript presented in an intelligible fashion and written in standard English?

Reviewer #1: Yes

Reviewer #2: Yes

Reviewer #3: Yes

5. Review Comments to the Author

Reviewer #1: COMMENTS TO THE AUTHORS

The manuscript ONE-D-21-36346, entitled “Distinction of Paramecium strains by a combination method of RAPD analysis and multiplex PCR” describes a method that was established as an useful strain identification using multiple Paramecium strains and to better identify isolates from the environment. Despite needing corrections, the manuscript presents interesting results which will contribute to the knowledge of the identification of Paramecium strains and the study of new species. In addition, english language should be revised. Thus, the manuscript is suitable for publication in the Plos One following major revision.

Abstract

Line 24: To specify which characteristics could be obtained and where they could be employed, as mentioned in the text.

Line 25: Replace “stablish” by “determine”.

Line 27: Delete “between” and replace “discriminate” by “differentiate”.

Line 28: Delete “interstrain”.

Line 30: Replace “kind” by “pattern”.

Line 31: Replace “settings” by “characteristics”.

Line 32: Replace “constructed” by “designed”.

Line 32: To specify how many primers were designed.

Line 33: Remove “Using this method”.

Line 33: Replace “A comparative” by “The comparative”.

Introduction

The introduction is too general with some confusing information. Some additional informations about the Chlorella, which are mentioned in the results, are necessary to give an overview.

Line 40-42: Replace “Paramecium is a single-cell, free-living eukaryote living in freshwaters, such as ponds and lakes, worldwide. It is also easy to cultivate and maintain artificially, and it is a model organism for many aspects of eukaryotic biology” by “Paramecium is a single-cell free-living eukaryote who lives in freshwaters worldwide, such as ponds and lakes. The artificially cultivation i salso easy, and it is considered a model organism for many aspects of eukaryotic biology”.

Line 46: Provide examples for the different symbiosis that are found for Paramecium, or make a link between this sentence and the following one about the P. bursaria.

Line 51: Replace “In the past” by “Erstwhile”.

Line 54: Replace “is” by “are”.

Line 55: Replace “accepted” by “studied”.

Line 55: To specify how many is related to “Relatively large number”.

Line 55: Replace “molecules” by “nucleic acids”.

Line 62: Explain why could be insufficient to distinguish.

Line 63: Explain what is a syngen.

Line 63: Add “related” before “the presence”.

Line 68: Remove “this”.

Line 69: Replace “any information on” by “previous details about”.

Line 70: Remove “target”.

Line 75: Replace “the” by “a”.

Line 77: Replace “weakness” by “gap”.

Line 81: Delete “new”.

Line 84: Replace “prepared” by “inserted”.

Material and Methods

Line 96: Replace “Applying” by “the DNA extraction”

Line 119: Replace “subjected” by “submitted”.

Results

The results are too general with some missing information. Authors are suggested to go deep in the results related to the figures, once they were not very approached.

Line 135: Replace “The NBRP Paramecium maintains ~1000 strains of 24 species. They designate 6 strains of P. caudatum, P. bursaria, and P. tetraurelia as standard strains and the other 53 strains of the 24 species as recommended strains” by “The NBRP designate 6 strains of P. caudatum, P. bursaria, and P. tetraurelia as standard strains and the other 53 strains of the 24 species as recommended strains for this study”.

Line 138-140: Delete “Genomic DNA was purified from the standard and recommended strains of P. caudatum, P. tetraurelia, and P. bursaria obtained from the NBRP, and RAPD analysis was performed using this DNA as a template” – this is material and methods, not results.

Line 140: To specify how many random primers.

Line 141: Remove “characteristic”.

Line 143: Identify in which figure is possible to notice the different pattern.

Line 148: P. caudatum in italics.

Line 149: Specify which are “these strains”.

Line 150: Authors elucidate that was slightly more difficult to distinguish the strains by the RAPD method. It is suggested to rewrite this part with a better overview over the figure 1. Furthermore, there are no specifications on the figure 1 about the Chrolella as mentioned in the text.

Line 153: P. bursaria with and without Chlorella are not clear neither in the text, nor in the figures. The authors are suggested to review the approach about the Chrolella. Otherwise, it may be deleted.

Line 159-166: Delete, once the sentences are not results.

Line 167: Delete “appearance”.

Line 171: Replace “because” by “once”.

Line 175: Specify which are the problems.

Discussion

In discussion, the authors are suggested to better discuss the figures regarding the importance about the results of them, and make a relation between the results found and the literature. For example, in the discussion the different patters found for each strain were not discussed, being of extreme importance. The authors are also suggested to provide an information about how many primers are needed to conclude which strain is related to the pattern found.

Line 188: Add a reference considering the “and identification of Paramecium strains based on this information are being carried out”

Line 190-192: Replace “Whole-genome sequencing is proceeding in several strains of Paramecium, but its utilization is still very limited. The cost, time, and simplicity of the procedure are important for the identification of strains” by “Whole-genome sequencing is proceeding in several strains of Paramecium, but its utilization is still very limited considering the cost and time.

Line 195-199: Replace “The RAPD method, which was also applied in this study, is a useful approach employed for a long time to identify and distinguish various species [20-22]. There are already several studies on the identification of Paramecium strains. However, one of the disadvantages that must be considered due to the properties of this method is the reproducibility of the results and the possibility that the quality of the template DNA will greatly affect the results” by “The RAPD method is a useful approach to identify and distinguish various species [20-22], including the identification of Paramecium strains. However, one of the disadvantages that must be considered is the reproducibility of the tests and the quality of the template DNA that may affect the results”.

Line 213: Add a reference in the end of the sentence.

References

The references may need some editing and corrections, authors should go through all the reference list to correct minor mistakes.

Figures

In all figures, authors are suggested to put the number of each sample in all three photos of the gels to facilitate the view. Besides that, a better way to link each reference strain and suggested strain would be good to make the understanding clear. The authors are also suggested to provide a final figure that would facilitate the overview of the results, summarizing all the finds of the study.

Reviewer #2: This study appears to have been conducted rigorously, and the results, particularly in terms of unique banding patterns, appear to support the authors' conclusions. However, it would be helpful for the reader if the authors delineated which banding patterns (which samples) were unique, and which parts of the patterns made a unique fingerprint in each figure for the RAPD analysis. This would help the reader determine for themselves which strains and primers would be useful for their own analyses. Also, how did the authors determine uniqueness of banding patterns? Was a program used, or was it just by eye? Having a standard rigorous method for this is key- it may have been covered in the previous paper, which was cited, but if so, it would be good to make it clear what they did for this study as well.

Also, it should be spelled out what NBRP is upon first usage of this acronym.

Reviewer #3: 1. The study presents the results of original research.

Yes

2. Results reported have not been published elsewhere.

No

3. Experiments, statistics, and other analyses are performed to a high technical standard and are described in sufficient detail.

Experiments have been described in detail, explicitly.

4. Conclusions are presented in an appropriate fashion and are supported by the data.

Conclusions have been appropriately described.

5. The article is presented in an intelligible fashion and is written in standard English.

The article has been presented in an adequate fashion and written in standard English

6. The research meets all applicable standards for the ethics of experimentation and research integrity.

The research meets all ethics of experimentation.

7. The article adheres to appropriate reporting guidelines and community standards for data availability.

The data availability statement does not appear in the manuscript.

6. PLOS authors have the option to publish the peer review history of their article (what does this mean?). If published, this will include your full peer review and any attached files.

Reviewer #1: No

Reviewer #2: **Yes: **Lydia Bright

Reviewer #3: **Yes: **ABDUL RAUF SHAKOORI

---

## [Author Response · Author response to Decision Letter 0]

3 Feb 2022

To Reviewer #1,

 Thank you for your valuable comments and suggestions. We have revised the manuscript according to these comments. Our responses to individual suggestions or questions are as follows.

>Line 24: To specify which characteristics could be obtained and where they could be employed, as mentioned in the text.

According to your suggestion, we have modified this sentence (line. 23-25).

>Line 25: Replace “stablish” by “determine”.

We have replaced the word according to your suggestion (line. 25).

>Line 27: Delete “between” and replace “discriminate” by “differentiate”.

We have modified the description according to your suggestion (line. 27).

>Line 28: Delete “interstrain”.

We have deleted the word according to your suggestion.

>Line 30: Replace “kind” by “pattern”.

We have replaced the word according to your suggestion (line. 31).

>Line 31: Replace “settings” by “characteristics”.

We have replaced the word according to your suggestion (line. 31).

>Line 32: Replace “constructed” by “designed”.

We have replaced the word according to your suggestion (line. 33).

>Line 32: To specify how many primers were designed.

We have designed 5 primer sets. We have added the description in the text (line. 32-33).

>Line 33: Remove “Using this method”.

We have deleted the word according to your suggestion.

>Line 33: Replace “A comparative” by “The comparative”.

We have replaced the word according to your suggestion (line. 34).

>The introduction is too general with some confusing information. Some additional information about the Chlorella, which are mentioned in the results, are necessary to give an overview.

We have modified the introduction according to your comments, including providing additional information about the Chlorella (line. 83-86). Please refer to our responses to your other specific comments or suggestions below.

>Line 40-42: Replace “Paramecium is a single-cell, free-living eukaryote living in freshwaters, such as ponds and lakes, worldwide. It is also easy to cultivate and maintain artificially, and it is a model organism for many aspects of eukaryotic biology” by “Paramecium is a single-cell free-living eukaryote who lives in freshwaters worldwide, such as ponds and lakes. The artificially cultivation is also easy, and it is considered a model organism for many aspects of eukaryotic biology”.

We have replaced the sentence according to your suggestion (line. 40-42).

>Line 46: Provide examples for the different symbiosis that are found for Paramecium, or make a link between this sentence and the following one about the P. bursaria.

We have added other example of symbiosis here (line. 46-48).

>Line 51: Replace “In the past” by “Erstwhile”.

We have replaced the word according to your suggestion (line. 53).

>Line 54: Replace “is” by “are”.

We have replaced the word according to your suggestion (line. 56).

>Line 55: Replace “accepted” by “studied”.

We have replaced the word according to your suggestion (line. 57).

>Line 55: To specify how many is related to “Relatively large number”.

This description has been removed for clarity of its meaning.

>Line 55: Replace “molecules” by “nucleic acids”.

This description has been removed for clarity of its meaning.

>Line 62: Explain why could be insufficient to distinguish.

We have modified the sentence (line. 60-63).

>Line 63: Explain what is a syngen.

We have added the brief explanation about syngen (line. 64-66).

>Line 63: Add “related” before “the presence”.

In relation to the above comment, the relevant text has been removed.

>Line 68: Remove “this”.

We have removed the word.

>Line 69: Replace “any information on” by “previous details about”.

We have replaced the word according to your suggestion (line. 72).

>Line 70: Remove “target”.

We have removed the word.

>Line 75: Replace “the” by “a”.

We have replaced the word (line. 78).

>Line 77: Replace “weakness” by “gap”.

We have replaced the word (line. 80).

>Line 81: Delete “new”.

We have removed the word.

>Line 84: Replace “prepared” by “inserted”.

We have replaced the word (line. 90).

>Line 96: Replace “Applying” by “the DNA extraction”

We have replaced the word according to your suggestion (line. 102).

>Line 119: Replace “subjected” by “submitted”.

We have replaced the word (line. 125).

>The results are too general with some missing information. Authors are suggested to go deep in the results related to the figures, once they were not very approached.

We have modified the results section according to your comments (line. 159-163 and 168-170). Please refer to our responses to your other specific comments or suggestions below.

>Line 135: Replace “The NBRP Paramecium maintains ~1000 strains of 24 species. They designate 6 strains of P. caudatum, P. bursaria, and P. tetraurelia as standard strains and the other 53 strains of the 24 species as recommended strains” by “The NBRP designate 6 strains of P. caudatum, P. bursaria, and P. tetraurelia as standard strains and the other 53 strains of the 24 species as recommended strains for this study”.

We have replaced the sentence (line. 144-145). The designation of standard strains and recommended strains by NBRP is not done for this study but for general user. Thus, “for this user” is not included in the revised manuscript.

>Line 138-140: Delete “Genomic DNA was purified from the standard and recommended strains of P. caudatum, P. tetraurelia, and P. bursaria obtained from the NBRP, and RAPD analysis was performed using this DNA as a template” – this is material and methods, not results.

We have deleted the sentence according to your suggestion.

>Line 140: To specify how many random primers.

We have described the number of primers used in this assay (line. 146).

>Line 141: Remove “characteristic”.

We have removed the word.

>Line 143: Identify in which figure is possible to notice the different pattern.

We have modified the description (line. 148-151). And we have added S1_Table summarizing the results.

>Line 148: P. caudatum in italics.

We have corrected the notation (line. 154).

>Line 149: Specify which are “these strains”.

We have modified the description to specify the meaning (line. 156).

>Line 150: Authors elucidate that was slightly more difficult to distinguish the strains by the RAPD method. It is suggested to rewrite this part with a better overview over the figure 1. Furthermore, there are no specifications on the figure 1 about the Chrolella as mentioned in the text.

We have added the description about Figure. 1 (line. 157-158).

And only P. bursaria maintains Chlorella cells among Paramecium species. Therefore, we have added the description about effects of the presence or absence of Chlorella on the results in Figure 3 (line. 159-163).

>Line 153: P. bursaria with and without Chlorella are not clear neither in the text, nor in the figures. The authors are suggested to review the approach about the Chrolella. Otherwise, it may be deleted.

Related to the other responses, we have added several statements about Chlorella (line. 159-163).

>Line 159-166: Delete, once the sentences are not results.

We have deleted the sentences or moved them from results section to methods section (line. 119-120, 138-139).

>Line 167: Delete “appearance”.

We have removed the word.

>Line 171: Replace “because” by “once”.

We have rephrased this sentence (line. 175-178) 

>Line 175: Specify which are the problems.

We have modified the description (line. 181-182).

>In discussion, the authors are suggested to better discuss the figures regarding the importance about the results of them, and make a relation between the results found and the literature. For example, in the discussion the different patters found for each strain were not discussed, being of extreme importance. The authors are also suggested to provide an information about how many primers are needed to conclude which strain is related to the pattern found.

According to your comments, we have revised the discussion and added some description (line. 204-208, 213-216, 254-255). We assume that it is difficult to identify Paramecium strains using a method that depends only on the RAPD analysis, and that it is important to use other approaches such as multiplex PCR used in this study in combination. We have also added supplemental table to provide an information about conclusion.

>Line 188: Add a reference considering the “and identification of Paramecium strains based on this information are being carried out”

We have added references here according to you suggestion (line. 194).

>Line 190-192: Replace “Whole-genome sequencing is proceeding in several strains of Paramecium, but its utilization is still very limited. The cost, time, and simplicity of the procedure are important for the identification of strains” by “Whole-genome sequencing is proceeding in several strains of Paramecium, but its utilization is still very limited considering the cost and time.

We have replaced the sentence (line. 196-198).

>Line 195-199: Replace “The RAPD method, which was also applied in this study, is a useful approach employed for a long time to identify and distinguish various species [20-22]. There are already several studies on the identification of Paramecium strains. However, one of the disadvantages that must be considered due to the properties of this method is the reproducibility of the results and the possibility that the quality of the template DNA will greatly affect the results” by “The RAPD method is a useful approach to identify and distinguish various species [20-22], including the identification of Paramecium strains. However, one of the disadvantages that must be considered is the reproducibility of the tests and the quality of the template DNA that may affect the results”.

We have replaced the sentence (line. 201-204).

>Line 213: Add a reference in the end of the sentence.

We have decided to add the supplemental data (S1_File) here.

>The references may need some editing and corrections, authors should go through all the reference list to correct minor mistakes.

We have checked the references again and editing them.

>In all figures, authors are suggested to put the number of each sample in all three photos of the gels to facilitate the view. Besides that, a better way to link each reference strain and suggested strain would be good to make the understanding clear. The authors are also suggested to provide a final figure that would facilitate the overview of the results, summarizing all the finds of the study.

According to your suggestion, we have modified all figures, especially the labeling of strains. And we have added a supplemental table (S1_Table) at the end of this paper to show the quick identification chart for each standard strain depending on the results of this study.

 

To Reviewer #2,

 Thank you for your valuable comments and suggestions. We have revised the manuscript according to these comments. Our responses to individual suggestions or questions are as follows.

>This study appears to have been conducted rigorously, and the results, particularly in terms of unique banding patterns, appear to support the authors' conclusions. However, it would be helpful for the reader if the authors delineated which banding patterns (which samples) were unique, and which parts of the patterns made a unique fingerprint in each figure for the RAPD analysis. This would help the reader determine for themselves which strains and primers would be useful for their own analyses. Also, how did the authors determine uniqueness of banding patterns? Was a program used, or was it just by eye? Having a standard rigorous method for this is key- it may have been covered in the previous paper, which was cited, but if so, it would be good to make it clear what they did for this study as well. Also, it should be spelled out what NBRP is upon first usage of this acronym. 

As you pointed out, it is extremely important to determine the band pattern in the PARD method. In this study, we did not use any some specific analysis programs or tools, but evaluated the band pattern by only appearance. We are fully aware of the significant effect of subjectivity on conclusions by this approach. Therefore, in this study, we have constructed a multiplex PCR method by applying the results of this RAPD, and have proposed a simple and stable method to distinguish between Paramecium strains by this method. In relation to this, a supplemental table (S1_Tabale) was added at the end of this paper to show the quick identification chart for each standard strain depending on the results of this study.

We also have added the full name of NBRP in the revised manuscript (line. 90).

 

To Reviewer #3,

 Thank you for your valuable comments. We have revised the manuscript according to these comments. Our response to individual comment is as follow.

>The data availability statement does not appear in the manuscript.

As for “Data Availability statement”, we have prepared a new S1 file. We state the following.

Data Availability: All relevant data are within the paper and its Supporting information file.

---

## [Decision Letter · Decision Letter 1]

24 Feb 2022

Distinction of Paramecium strains by a combination method of RAPD analysis and multiplex PCR

PONE-D-21-36346R1

Dear Dr. Watarai,

We’re pleased to inform you that your manuscript has been judged scientifically suitable for publication and will be formally accepted for publication once it meets all outstanding technical requirements.

Kind regards,

Maria Stefania Latrofa

Academic Editor

PLOS ONE

Additional Editor Comments:

Dear Authors, the article has been improved and is accepted for publication,

I would suggest only few modification as mentioned below:

line 33: change in “5 specific primer sets  were designed and a multiplex PCR method was developed”

line 36: change in “specific strains were identified”; specify “This method”, which one?

line 48: write the name “P” in full

line 55 and through the text change “18S rDNA” in “18S rRNA” and put “COI” I lower letter, the capital are used for protein product

line 66: change “and in some cases are recognized as a cryptic species”

lines 90-91: change in “were used, and a novel PCR tool,  was applied for strains identification”

Lines 145-148: Change in “After the comparative study using these designated strains and ten random primer sets from a previous study   [24, 26, 27], differences in band patterns were found between strains using only the random primer-02, -03, and -05 (Figs. 1–3)”

Lines 153-154: remove the sentence “These strains could be identified only by the results of these RAPD analysis”, is superfluous

Reviewers' comments:

Reviewer's Responses to Questions

**Comments to the Author**

1. If the authors have adequately addressed your comments raised in a previous round of review and you feel that this manuscript is now acceptable for publication, you may indicate that here to bypass the “Comments to the Author” section, enter your conflict of interest statement in the “Confidential to Editor” section, and submit your "Accept" recommendation.

Reviewer #1: All comments have been addressed

2. Is the manuscript technically sound, and do the data support the conclusions?

Reviewer #1: Yes

3. Has the statistical analysis been performed appropriately and rigorously? 

Reviewer #1: N/A

4. Have the authors made all data underlying the findings in their manuscript fully available?

Reviewer #1: Yes

5. Is the manuscript presented in an intelligible fashion and written in standard English?

Reviewer #1: Yes

6. Review Comments to the Author

Reviewer #1: The manuscript ONE-D-21-36346, entitled ''Distinction of Paramecium strains by a combination method of RAPD analysis and multiplex PCR'' was correctly reviewed by the authors that reinforced the interesting results that highlighted the knowledge of the study. With the new corrections, the article is now acceptable.

7. PLOS authors have the option to publish the peer review history of their article (what does this mean?). If published, this will include your full peer review and any attached files.

Reviewer #1: **Yes: **Viviane Noll Louzada Flores

---

## [Editor Report · Acceptance letter]

4 Mar 2022

PONE-D-21-36346R1 

Distinction of *Paramecium* strains by a combination method of RAPD analysis and multiplex PCR 

Dear Dr. Watarai:

I'm pleased to inform you that your manuscript has been deemed suitable for publication in PLOS ONE. Congratulations! Your manuscript is now with our production department. 

Kind regards, 

on behalf of

Dr. Maria Stefania Latrofa 

Academic Editor

PLOS ONE